# The Role of Macrophages in Connective Tissue Disease-Associated Interstitial Lung Disease: Focusing on Molecular Mechanisms and Potential Treatment Strategies

**DOI:** 10.3390/ijms241511995

**Published:** 2023-07-26

**Authors:** Chia-Chun Tseng, Ya-Wen Sung, Kuan-Yu Chen, Pin-Yi Wang, Chang-Yi Yen, Wan-Yu Sung, Cheng-Chin Wu, Tsan-Teng Ou, Wen-Chan Tsai, Wei-Ting Liao, Chung-Jen Chen, Su-Chen Lee, Shun-Jen Chang, Jeng-Hsien Yen

**Affiliations:** 1Graduate Institute of Clinical Medicine, College of Medicine, Kaohsiung Medical University, Kaohsiung 807, Taiwan; 2School of Medicine, College of Medicine, Kaohsiung Medical University, Kaohsiung 807, Taiwan; 3Division of Rheumatology, Department of Internal Medicine, Kaohsiung Medical University Hospital, Kaohsiung 807, Taiwan; 4Department of Nursing, Kaohsiung Medical University Hospital, Kaohsiung 807, Taiwan; 5Department of Biotechnology, College of Life Science, Kaohsiung Medical University, Kaohsiung 807, Taiwan; 6Department of Medical Research, Kaohsiung Medical University Hospital, Kaohsiung 807, Taiwan; 7Department of Internal Medicine, Kaohsiung Municipal Ta-Tung Hospital, Kaohsiung 801, Taiwan; 8Laboratory Diagnosis of Medicine, College of Medicine, Kaohsiung Medical University, Kaohsiung 807, Taiwan; 9Department of Kinesiology, Health and Leisure Studies, National University of Kaohsiung, Kaohsiung 811, Taiwan; 10Institute of Biomedical Sciences, National Sun Yat-Sen University, Kaohsiung 804, Taiwan

**Keywords:** connective tissue disease, interstitial lung disease, macrophage, systemic sclerosis

## Abstract

Connective tissue disease-associated interstitial lung disease (CTD-ILD) is a severe manifestation of CTD that leads to significant morbidity and mortality. Clinically, ILD can occur in diverse CTDs. Pathologically, CTD-ILD is characterized by various histologic patterns, such as nonspecific interstitial pneumonia, organizing pneumonia, and usual interstitial pneumonia. Abnormal immune system responses have traditionally been instrumental in its pathophysiology, and various changes in immune cells have been described, especially in macrophages. This article first briefly overviews the epidemiology, clinical characteristics, impacts, and histopathologic changes associated with CTD-ILD. Next, it summarizes the roles of various signaling pathways in macrophages or products of macrophages in ILD, helped by insights gained from animal models. In the following sections, this review returns to studies of macrophages in CTD-ILD in humans for an overall picture of the current understanding. Finally, we direct attention to potential therapies targeting macrophages in CTD-ILD in investigation or in clinical trials, as well as the future directions regarding macrophages in the context of CTD-ILD. Although the field of macrophages in CTD-ILD is still in its infancy, several lines of evidence suggest the potential of this area.

## 1. Introduction

Interstitial lung diseases (ILDs) comprise a heterogeneous group of multiple entities characterized by damage to the lung parenchyma due to varying degrees of inflammation and fibrosis [1]. ILDs arise from a broad spectrum of known or unknown etiologies and inflict a substantial burden on patients and health care systems [2]. Connective tissue disease (CTD), also known as collagen vascular disease [3], is a common cause associated with ILD [4], with 15% of ILDs arising from CTD [5]. CTD includes several disease entities, such as dermatomyositis (DM), microscopic polyangiitis (MPA), rheumatoid arthritis (RA), and systemic sclerosis (SSc). Corresponding to CTD-associated ILDs (CTD-ILDs) are DM-associated ILD (DM-ILD), MPA-associated ILD (MPA-ILD), RA-associated ILD (RA-ILD), and SSc-associated ILD (SSc-ILD). Pathologically, CTD-ILD can be classified based on respective histopathologic patterns, including nonspecific interstitial pneumonia (NSIP), organizing pneumonia (OP), and usual interstitial pneumonia (UIP) [6]. In practice, CTD-ILD is usually diagnosed through multidisciplinary discussion (MDD) including review of the clinical features, imaging, serology, and pathology by pulmonologists, radiologists, rheumatologists, and pathologists [7].

Clinically, different CTD-ILDs have distinct epidemiologies, pathologic changes, comorbidities, and prognoses (Figure 1). The incidence of DM-ILD is 1011/10^5^ [8]; it is characterized by OP, NSIP, and UIP [9] and associated with palmar papules and myocardial involvement [10,11], and its mortality rate is as high as 24.62% [12]. The estimated incidence of MPA-ILD is 1.4/10^5^ [13]; it is characterized by UIP and NSIP [14] and associated with decreased renal function [15], and its mortality rate is up to 39% [16]. The estimated incidence of RA-ILD is 3.7/10^5^ [17]; it is presented as UIP [18] and associated with increased asthma, chronic obstructive pulmonary disease (COPD), diabetes, and heart disease [19,20], and its mortality rate is 8.08% [17]. The incidence of SSc-ILD is 1364/10^5^ [21]; it is characterized by NSIP and UIP [14] and is associated with COPD, gastroesophageal reflux disease (GERD), hypertension, and skin disorders [22], and it has high mortality rate (39%) [23]. Overall, in CTD, the presence of ILD could lead to increased morbidity and mortality [24].

Although the mechanisms contributing to ILD in the context of CTDs are not known in detail, the prevailing hypothesis suggests the involvement of abnormal inflammation and subsequent exaggerated fibrotic reactions [1]. Interestingly, macrophages not only coordinate inflammation process but also mediate the formation of fibrosis. Given these characteristics of macrophages, it makes sense that macrophages play critical roles in the pathogenesis of CTD-ILD. The roles of macrophages have been suggested in numerous studies. For example, macrophages in the lungs of SSc-ILD upregulated profibrotic factors that promoted lung fibroblast activation [25]. Another example is DM-ILD. Previous studies reported that ferritin was produced by macrophages [26] and correlated with lung function parameters and survival [27]. IL-18 was also secreted by macrophages [28], elevated in DM-ILD compared with DM without ILD [29], and correlated with the severity of pulmonary disease [30]. Together, these studies point to the importance of macrophages in the development and progression of CTD-ILD.

Macrophages are versatile cells that exhibit high plasticity according to crosstalk between macrophages and the tissue microenvironment. Historically, analogous to the Th1/Th2 dichotomy, studies have characterized macrophages as classically activated (M1) or alternatively activated (M2) according to the activation status. Although the dichotomous inflammatory M1/repair M2 model cannot fully represent the current understanding of macrophage biology in ILD, studies on fibrosis usually build upon this classification scheme. As one of the most versatile cell types with diverse immunity functions, macrophages not only initiate an inflammatory response after injury but are also involved in the injury resolution and repair. In other words, in the airway and lung microenvironment, macrophages orchestrate the development and establishment of ILD. In this process, macrophage products and/or macrophages themselves are implicated at each key step. Work utilizing ILD patient samples provided several pieces of evidence of altered macrophages in ILD, whose functional importance was further confirmed in murine models [31].

Radiologically, CTD-ILD is characterized by various patterns of fibrosis using high-resolution computed tomography (HRCT) in the lung bases of patients with CTD, such as RA and SSc [32]. Most of our understanding of CTD-ILD comes from studies on pulmonary fibrosis (PF) in animal models. Therefore, we will first review the current knowledge of macrophages in the ILD/PF of animal models and then review recent advances in the molecular biology of CTD-ILD in the literature.

## 2. Roles of Signaling Pathways of Macrophages in ILD/PF in Animal Models

Regarding the relationship between macrophages and PF, several studies support the vital roles of macrophages in the pathogenesis of lung fibrosis. Macrophages in lung tissue samples from patients with fibrosis are higher compared to controls without fibrosis [33]. Macrophages play a mechanistic role in exacerbating lung fibrosis [34]. On the other hand, the ablation of macrophages was shown to significantly decrease fibroblasts and collagen deposition, ameliorating lung fibrosis [35]. Macrophages participate in both the inflammation and tissue repair phases of PF through various mechanisms, as detailed below.

(1) Caspase pathway: Studies show that MCU in PF macrophages leads to augmented expression of carnitine palmitoyltransferase 1A (CPT1A). By inducing Bcl-2 expression, CPT1A decreases caspase-3 activity, which attenuates macrophage apoptosis and thereby facilitates lung fibrosis progression (Table 1). The deletion of Bcl-2 in macrophages protects mice from developing PF [36].

(2) CCR2 axis: The CCL2/CCR2 axis is a major regulator of monocyte trafficking and plays an essential role in PF. CCR2 deficiency suppresses macrophage infiltration and reduces macrophage-derived MMP-2 and MMP-9 production, which decreases lung extracellular matrix content in the lungs (Table 1) and protects mice from PF [37].

(3) CCR4 axis: CCR4 promotes tissue injury through the induction of the M1 macrophage phenotype. In the absence of CCR4, macrophages upregulate the scavenging receptor D6, which attenuates inflammation and tissue injury (Table 1) and protects mice from lung fibrosis [38].

(4) CD204 axis: Collagen type I monomers stimulate macrophages to induce CD204 expression. In the context of elevated CD204, PF macrophages show hyperreactivity to stimulation with collagen type I monomers, resulting in exacerbated CCL18 secretion [39]. CCL18 triggers collagen production by fibroblasts. However, through β_2_-integrins and scavenger receptors, macrophages in the vicinity bind to collagen type I and increase CCL18 production. These reactions generate a feed-forward loop of augmented, ceaseless macrophage activation and unrestricted collagen production by fibroblasts (Table 1) [40].

(5) Glycolysis: Macrophages from fibrotic lungs assume the elevated expression of the glucose transporter GLUT1 [41], which augments the glycolysis required for the profibrotic profile in macrophages from fibrotic lungs (Table 1) [42].

(6) GSK pathway: TRIB3 is significantly upregulated in the macrophages of patients with PF. The TRIB3-GSK-3β interaction inhibits A20 activity and stabilizes C/EBPβ to induce macrophage activation, which triggers the transformation of lung fibroblasts into myofibroblasts, driving lung fibrosis (Table 1). Consequently, the macrophage-specific knockout of *TRIB3* suppresses fibrotic changes in the lungs [43].

(7) HIF pathway: Macrophages of fibrotic lungs show increased HIF1A, which upregulates the ADORA2B receptor on macrophages. ADORA2B contributes to macrophage differentiation and the production of profibrotic mediators (Table 1), facilitating fibrosis in the lung [44].

(8) Itaconate axis: In the murine PF model, aconitate decarboxylase 1 (ACOD1) decarboxylates cis-aconitate to itaconate, and itaconate suppresses fibroblast proliferation and profibrotic activity, thereby limiting the severity of PF (Table 1). *ACOD1* deficiency in macrophages induces profibrotic gene expression and worsens lung fibrosis [45].

(9) Macrophage migration: In the process of lung fibrosis, macrophages migrate into or within the lung to orchestrate and amplify fibrosis. In humans, monocyte migration is enhanced in PF. PLXNC1 is underexpressed and Syt7 is overexpressed in PF, and PLXNC1 suppresses Syt7-driven macrophage migration (Table 1). The underexpression or absence of PLXNC1 exacerbates macrophage migration and aggravates experimentally induced lung fibrosis. Conversely, restoring PLXNC1 in macrophages is sufficient to attenuate fibrosis [46].

(10) MAPK signaling: Human fibrotic lung exhibits augmented FOXM1 expression in macrophages. FOXM1 activates DUSP1 and inhibits the p38 MAPK signaling, therefore ameliorating lung fibrosis (Table 1). In agreement with this, macrophage-specific FOXM1 knockout mice develop severe PF. On the other hand, the transfer of FOXM1-expressing monocytes protects *FOXM1*-deficient mice against lung fibrosis [47].

(11) STAT6 signaling: Cu,Zn-superoxide dismutase (Cu,Zn-SOD) is expressed in macrophages [48]. Cu,Zn-SOD-mediated H_2_O_2_ generation activates STAT6 (Table 1) [49]. In the same way, macrophages isolated from fibrotic lungs have elevated Gab2, which increases STAT6 activation (Table 1) [50]. SART1 in macrophages facilitates the STAT6 signaling axis (Table 1) [51]. S1PR2 increases STAT6 activation (Table 1) [52]. These changes all accelerate STAT6 signaling and subsequent M2 polarization, triggering the development of PF [49,50,51,52].

(12) TGF-β signaling: There are several ways to modulate macrophage TGF-β signaling in PF. For example, Akt-mediated reactive oxygen species (ROS) production induces mitophagy and contributes to macrophage apoptosis resistance, which results in upregulated transforming growth factor-β (TGF-β) [53]. Similarly, C/EBP homologous protein (CHOP) decreases the expression of SOCS1 and SOCS3, thereby enhancing STAT6/PPARγ signaling, which is essential for TGF-β production (Table 1) [54]. Another example is MBD2. MBD2 selectively binds to the *SH2-containing inositol 5′-phosphatase* (*SHIP*) promoter in macrophages, by which MBD2 represses *SHIP* expression and enhances PI3K/Akt signaling to promote the macrophage M2 program and the production of downstream TGF-β [55]. In the same way, *FBXW7* is an E3 ubiquitin ligase, whose expression in the macrophages of pulmonary tissue fibrosis mice is markedly decreased, and the deficiency of macrophage *FBXW7* promotes the recruitment and accumulation of phagocytes, increases the K48-linked polyubiquitination and proteasome degradation of c-Jun, and downregulates the expression of TGF-β (Table 1) [56]. When TGF-β is increased through these pathways, enhanced fibroblast differentiation and fibrosis ensues [53].

(13) TNF pathway: STAT1 is expressed in lung macrophages [57]. STAT1 induces ICAM-1 and downstream TNF from macrophages and the subsequent infiltration of inflammatory cells and eventual fibrosis (Table 1) [58]. In agreement with this, STAT1 inhibition ameliorates collagen deposition in the lungs in an animal model [59]. In the same way, CD300c2 enhances high-mobility group box protein-1 (HMGB-1)-induced macrophage activation to produce tumor necrosis factor (TNF), which is a leukocyte chemoattractants, resulting in the accumulation of augmented immune cells, inflammation, and the aggravation of lung fibrosis (Table 1) [60].

**Table 1 ijms-24-11995-t001:** Signaling pathways of macrophages in ILD/PF in animal models. “🡪”—lead to, “↓”—decrease, “↑”—increase.

Signaling Pathways	Results	Reference
Caspase pathway		
MCU ↑ CPT1A 🡪 ↑ Bcl-2 🡪 ↓ caspase-3 activity	↓ macrophage apoptosis 🡪 ↑ fibrosis	[36]
CCR2 axis		
CCR2 ↑ macrophage infiltration 🡪 ↑ MMP-2/MMP-9	↑ extracellular matrix deposition	[37]
CCR4 axis		
CCR4 ↑ inflammation and tissue injury	↑ fibrosis	[38]
CD204 axis		
collagen type I monomers ↑ CD204	↑ CCL18 🡪 ↑ collagen production 🡪 ↑ CCL18	[39,40]
Glycolysis		
GLUT1 ↑ glycolysis	↑ profibrotic macrophages	[42]
GSK pathway		
TRIB3-GSK-3β interaction ↓ A20 activity and stabilizes C/EBPβ 🡪 macrophage activation	↑ fibroblasts transform to myofibroblasts	[43]
HIF pathway		
HIF1A ↑ ADORA2B 🡪 ↑ macrophage differentiation	↑ profibrotic mediators	[44]
Itaconate axis		
ACOD1 🡪 ↑ itaconate	↓ fibroblast proliferation and profibrotic activity	[45]
Macrophage migration		
PLXNC1	↓ macrophage migration	[46]
MAPK signaling		
FOXM1 activates DUSP1 🡪 ↓ p38 MAPK signaling	↓ fibrosis	[47]
STAT6 signaling		
Cu,Zn-SOD ↑ H_2_O_2_ 🡪 activates STAT6	↑ M2 polarization	[49]
Gab2 🡪 activates STAT6		[50]
SART1 🡪 ↑ activates STAT6		[51]
S1PR2 🡪 ↑ activates STAT6		[52]
TGF-β signaling		
Akt ↑ ROS 🡪 ↑ mitophagy ↑ apoptosis resistance 🡪 ↑ TGF-β	↑ fibroblast differentiation	[53]
CHOP ↓ SOCS1/SOCS3 🡪 ↑ STAT6/PPARγ signaling 🡪 ↑ TGF-β		[54]
MBD2 ↓ *SHIP* 🡪 ↑ PI3K/Akt signaling 🡪 ↑ M2 polarization 🡪 ↑ TGF-β		[55]
*FBXW7* ↓ phagocyte recruitment ↑ c-Jun degradation 🡪 ↓ *TGF-*β		[56]
TNF pathway		
STAT1 🡪 ↑ ICAM-1 🡪 ↑ TNF	↑ inflammatory cells infiltration 🡪 ↑ fibrosis	[58,59]
CD300c2 🡪 ↑ macrophage activation🡪 ↑ TNF		[60]

## 3. Roles of Macrophage-Derived Secretory Proteins/microRNAs in ILD/PF in Animal Models

In addition to macrophages themselves, macrophage-derived secretory proteins and microRNAs also execute various biological functions to modulate the lung fibrotic process.

### 3.1. Macrophage-Derived Secretory Proteins/microRNAs That Aggravate ILD/PF in Animal Models

(1) ADAM17: ADAM17 in association with membrane-bound IL-6Rα (mIL-6Rα) is increased in macrophages in fibrotic lungs. ADAM17 promotes the shedding of mIL-6Rα from the membranes of activated macrophages, thereby initiating IL-6 trans-signaling. IL-6 trans-signaling enhances fibroblast proliferation and extracellular matrix (ECM) production (Table 2) and the neutralization of IL-6 trans-signaling attenuates PF [61].

(2) AT1R: Macrophages secrete a considerable number of exosomes bearing AT1R, which are taken up by fibroblasts and result in higher levels of AT1R, the activation of the profibrotic TGF-β/Smad2/Smad3 pathway, the production of α-collagen I, and augmented Ang II secretion by fibroblasts. Interestingly, Ang II increases the number of macrophage exosomes and AT1R secretion, leading to a positive feedback between Ang II and exosome production involved in lung fibrosis (Table 2) [62].

(3) CCL6: CCL6, also called C10, is expressed in the macrophages and also attracts macrophages. Upregulated CCL6 accounts for the increased susceptibility to PF in the mice model. The neutralization of CCL6 attenuates subsequent lung fibrosis (Table 2) [63].

(4) Fibronectin: Compared to normal macrophages, PF macrophages produce more fibronectin [64]. Fibronectin transform fibroblasts to myofibroblasts, which lead to a local accumulation of extracellular matrix and hence the development of fibrosis in the lungs (Table 2) [65].

(5) Galectin-3: Galectin-3 is produced by macrophages and the levels of galectin-3 in CTD-ILD patients are higher than in control patients. Galectin-3 induces the production of TNF-α in macrophages, and galectin-3 expression in macrophages is also induced by TNF-α, creating a self-perpetuating cycle between TNF-α and galectin-3 to support inflammation. In addition, galectin-3 stimulates fibroblasts to induce proliferation (Table 2). All these changes promote lung fibrosis [66].

(6) IGF-I: Macrophages secrete IGF-I, which is increased in PF patients. Macrophage-derived IGF-I activates the prosurvival kinase Akt, which protects myofibroblasts from apoptosis, contributing to fibrosis (Table 2) [67]. Correspondingly, decreasing IGF-I ameliorates the severity of lung fibrosis [37].

(7) IL-1β: IL-1β induces inflammation with leukocyte influx and upregulates myofibroblast differentiation, resulting in augmented pulmonary collagen deposition (Table 2) and lung fibrosis [68,69].

(8) MIP-1α: Macrophages secrete MIP-1α, which promotes leukocyte accumulation [70]. On the contrary, anti-MIP-1α antibodies significantly reduce pulmonary mononuclear phagocyte accumulation and ensue fibrosis (Table 2) [37].

(9) miR-328: Macrophage-derived miR-328 inhibits the family with sequence similarity 13 member A (FAM13A). This induces lung fibroblast proliferation and the overexpression of collagen 1A, collagen 3A, and alpha-smooth muscle actin (α-SMA), which aggravate fibrosis (Table 2) [71]. On the other hand, silencing miR-328 in M2 macrophages alleviates the progression of lung tissue fibrosis [72].

(10) NTN1: NTN1^+^ macrophages also accumulate in fibrotic lungs. Through interaction with DCC, macrophage-derived NTN1 impacts adrenergic nerve remodeling, which is required for lung fibrosis (Table 2) [73].

(11) PDGF: Macrophages from animal models of PF secrete PDGF, which is a potent mitogen for fibroblasts, facilitating lung fibrosis (Table 2) [74].

(12) S100A4: M2-polarized macrophages secrete S100A4, which stimulates the proliferation and activation of fibroblasts and is increased in PF (Table 2). The inhibition of S100A4 reduces histological evidence of lung fibrosis [75].

(13) TGF-β: In the PF model, macrophages secrete TGF-β [76]. TGF-β increases PAI-1 expression in macrophages via Smad3, leading to extracellular stroma accumulation. These abnormalities favor lung fibrosis [77].

(14) Wnt: Wnt expression is noted in macrophages [78]. Wnt/β-catenin signaling promotes the differentiation of infiltrated monocyte-macrophage populations that contribute to lung fibrosis (Table 2). Therefore, the loss of the Wnt coreceptor low density lipoprotein receptor-related protein 5 (LRP5) attenuates the lung fibrotic process [79].

**Table 2 ijms-24-11995-t002:** Macrophage-derived secretory proteins/microRNAs in ILD/PF in animal models. “🡪”—lead to, “↓”—decrease, “↑”—increase.

Molecules	Characteristics	Signaling Pathway	Reference
Macrophage-derived secretory proteins/microRNAs that aggravate ILD/PF			
ADAM17	Protein	↑ shedding of mIL-6Rα 🡪 ↑ IL-6 trans-signaling 🡪 ↑ fibroblast proliferation and ECM production	[61]
AT1R	Protein	AT1R taken up by fibroblasts 🡪 ↑ profibrotic ↑ TGF-β/Smad2/Smad3 pathway, ↑ α-collagen I production, ↑ Ang II secretion by fibroblasts 🡪 ↑ macrophage exosomes and AT1R in exosomes	[62]
CCL6	Protein	↑ fibrosis	[63]
Fibronectin	Protein	↑ myofibroblasts 🡪 ↑ extracellular matrix deposition	[64,65]
Galectin-3	Protein	↑ TNF-α 🡪 ↑ galectin-3 🡪 ↑ inflammation, ↑ fibroblast proliferation	[66]
IGF-I	Protein	activates Akt 🡪 ↓ myofibroblast apoptosis	[67]
IL-1β	Protein	↑ leukocyte influx, ↑ myofibroblasts 🡪 ↑ collagen deposition	[68,69]
MIP-1α	Protein	↑ leukocyte accumulation 🡪 ↑ fibrosis	[37,70]
miR-328	microRNA	↓ FAM13A 🡪 ↑ fibroblast proliferation and ↑ collagen 1A, ↑ collagen 3A, ↑ α-SMA	[71]
NTN1	Protein	interacts with DCC 🡪 impacts adrenergic nerve remodeling 🡪 ↑ fibrosis	[73]
PDGF	Protein	a mitogen for fibroblasts	[74]
S100A4	Protein	↑ fibroblast proliferation and activation	[75]
TGF-β	Protein	↑ PAI-1 🡪 ↑ extracellular matrix	[77]
Wnt	Protein	↑ differentiation of monocyte-macrophage 🡪 ↑ fibrosis	[79]
Macrophage-derived secretory proteins/microRNAs that ameliorate ILD/PF			
Pentraxin-2 (serum amyloid P) [80]	Protein	↓ M2 differentiation	[81]
miR-142-3p	microRNA	↓ TGFβ-R1 🡪 ↓ profibrotic genes	[82]

### 3.2. Macrophage-Derived Secretory Proteins/microRNAs That Ameliorate ILD/PF in Animal Models

(1) Pentraxin-2: Pentraxin-2, also called serum amyloid P (SAP) [80], reduces M2 differentiation and subsequent TGF-β-mediated lung fibrosis (Table 2) [81].

(2) miR-142-3p: Macrophages secrete antifibrotic miR-142-3p, which reduces the expression of transforming growth factor β receptor 1 (TGFβ-R1) and downstream profibrotic genes in PF (Table 2) [82].

## 4. Roles of Macrophages in CTD-ILD

The evidence mentioned above suggests that macrophages actively participate in the pathogenesis of ILD/PF. In theory, macrophages could also analogously take part in the course of CTD-ILD. This is supported by several observations.

### 4.1. Macrophages in Dermatomyositis (DM)-Associated ILD (DM-ILD)

Although there are no animal models that directly address the molecular mechanisms through which macrophages contribute to DM-ILD, several studies provide indirect evidence implicating macrophages as the main player in DM-ILD. Pathological analyses show that the infiltration of CD163-positive macrophages into the alveolar spaces is more severe in fatal DM-ILD than in DM-ILD survivors [83], and increased serum CD163 levels are associated with a higher mortality rate in DM-ILD [84]. Similarly, CD206-positive macrophages accumulate more densely in fatal cases of DM-ILD than in DM-ILD survivors, and elevated serum CD206 levels are associated with a higher mortality rate in DM-ILD [84]. Additionally, macrophages express resistin, which is significantly higher in DM-ILD than in DM patients without ILD. Compared with chronic ILD, resistin levels are significantly elevated in rapidly progressive ILD. Acting via nuclear factor kappa B (NFκB) signaling, resistin upregulates IL-1, IL-6, and TNF-α in human monocytes. These proinflammatory cytokines induce resistin expression in macrophages (Figure 2), triggering a self-sustained loop for lung injury [85]. Together, these studies highlight the contribution of macrophages to DM-ILD.

### 4.2. Macrophages in Microscopic Polyangiitis (MPA)-Associated ILD (MPA-ILD)

In MPA-ILD, macrophage polarization is skewed toward M2, which recruits fibroblasts and transforms them into myofibroblasts, culminating in the formation of lung fibrosis [86]. In the process of M2 polarization, the infiltrated macrophages produce CCL2, which recruits more macrophages to infiltrate the alveolar spaces. The recruited macrophages secrete CCL2, promoting a vicious cycle (Figure 2). In addition, in concert with PDGF secreted by vascular endothelial cells, CCL2 promotes myofibroblast differentiation and ECM production (Figure 2), leading to lung fibrosis [87]. Collectively, these findings implicate macrophages in the pathogenesis of MPA-ILD.

### 4.3. Macrophages in Rheumatoid Arthritis (RA)-Associated ILD (RA-ILD)

In RA-ILD, autoimmune/complement/interferon cascade genes are altered in macrophages, which might contribute to profibrotic inflammatory lung responses [88]. Several molecules produced by macrophages have been highlighted in the literature (Figure 2).

In RA-ILD, macrophages produce higher IL-6, which promotes the expansion of GM-CSF-producing T cells. GM-CSF is a strong inducer of neutrophil infiltration into the lung and contributes to the progression of RA-ILD in an animal model (Figure 2) [89]. SDC2 in macrophages activates CD148 in fibroblasts, inhibits PI3K/Akt signaling, downregulates PAD2 in fibroblasts, and attenuates collagen production by fibroblasts (Figure 2). The overexpression of SDC2 in alveolar macrophages decreases collagen deposition in the lungs and protects mice from RA-ILD [90]. The augmented secretion of TNF by alveolar macrophages from RA-ILD has been noted [91]. TNF induces an inflammatory phase that is predominated by cellular infiltration to the pulmonary tissue, which subsequently transitions to a fibrotic phase that constitutes the “irreversible” process of collagen deposition in the pulmonary parenchyma, resulting in RA-ILD in a murine model (Figure 2) [92].

### 4.4. Macrophages in Systemic Sclerosis (SSc)-Associated ILD (SSc-ILD)

In SSc-ILD, macrophages express profibrotic factors that promote the differentiation, migration, and activation of fibroblasts, suggesting the involvement of macrophages in SSc-ILD [25]. Currently, there are several known mechanisms that macrophages utilize to modulate SSc-ILD (Figure 2). (1) Monocytes from SSc-ILD patients reveal a profibrotic phenotype characterized by the expression of CD163 and the enhanced secretion of CCL18 [93], which promotes collagen production (Figure 2) [40]. (2) At the same time, SSc-ILD macrophages release significantly more fibronectin, which leads to fibroblast proliferation and subsequent collagen deposition (Figure 2) [64,93]. (3) Immunologically, immune complexes (ICs) activate human monocytes to secrete M-CSF and IL-6, which in turn induce osteopontin (OPN) from monocytes. Next, OPN facilitates fibroblast migration to areas adjacent to the fibrotic niche, thereby aiding lung fibrosis progression (Figure 2) [94]. (4) Meanwhile, PLXNC1 expression is reduced in SSc-ILD macrophages, which could lead to the excess migration of macrophages (Figure 2), amplifying fibrosis [46]. (5) Moreover, the response gene to complement 32 (RGC32) is abundantly expressed in macrophages, activates NFκB signaling, and promotes inflammatory gene expression by binding to their promoters. *RGC32* deficiency in mice impairs the polarization of classically activated macrophages, attenuates the expression of inflammatory mediators in macrophages, including the fibrosis inducers iNOS and IL-1β, which are regulated by NFκB, and significantly ameliorates lung fibrosis (Figure 2) [95]. (6) YKL-40 is elevated in SSc-ILD. Macrophages produce YKL-40, which exerts promitogenic effects on lung fibroblasts (Figure 2) [96]. These findings all support the central roles of macrophages in SSc-ILD.

## 5. Targeting Macrophages in CTD-ILD

It is hoped that, apart from detailing our current understanding of CTD-ILD pathogenesis, this knowledge could be leveraged to create therapeutic strategies to combat CTD-ILD. However, similar to the understanding of disease pathogenesis, most therapeutic strategies that target macrophages in CTD-ILD originated from studies targeting macrophages in PF. Therefore, in the following section, we review therapeutic strategies to target macrophages in preclinical or clinical models of PF or CTD-ILD (Figure 3):

(1) ABT-199: ABT-199, an inhibitor of Bcl-2, is a potential therapeutic agent that prevents fibrosis progression. Bcl-2 decreases caspase-3 activity, mediating the apoptosis resistance of macrophages, which facilitates lung fibrosis (Figure 3). Mice have lung fibrosis resolution when treated with ABT-199 after fibrosis is established [36].

(2) Anti-IL-33: IL-33 is induced in macrophages during PF. Acting through ST2, IL-33 polarizes M2 macrophages to increase IL-13 and TGF-β1 production. This activates fibroblasts and stimulates fibroblast proliferation. Therefore, IL-33 enhances collagen synthesis by fibroblasts and amplifies the fibrosis of the lung. Accordingly, an anti-IL-33 antibody reduces lung fibrosis (Figure 3) [97].

(3) Clevudine: Clevudine is a purine nucleoside analog indicated for hepatitis B virus (HBV) treatment. Interestingly, clevudine can inhibit the PI3K/Akt signaling and therefore suppresses M2 differentiation. Correspondingly, epithelial-to-mesenchymal transition (EMT) and M2-induced myofibroblast activation are alleviated (Figure 3), thus preventing profibrotic cytokine secretion and downstream collagen deposition, with a concomitant attenuation of fibrotic processes in the lungs [98].

(4) Leflunomide: Leflunomide inhibits Cu,Zn-SOD-mediated H_2_O_2_ generation. Consequently, Jmjd3 expression is reduced, preventing macrophage M2 polarization and thereby attenuating lung tissue fibrosis (Figure 3) [49].

(5) GW4869: Macrophages secrete exosomes carrying AT1R, which are taken up by fibroblasts, activating collagen synthesis. GW4869, by inhibiting exosome secretion, attenuates fibrosis in a mouse model (Figure 3) [62].

(6) IL-10: IL-10 suppresses TGF-β production in macrophages and subsequently decreases collagen production in fibroblasts (Figure 3), diminishing the intensity of pulmonary tissue fibrosis [99].

(7) Methyl palmitate: Methyl palmitate inhibits the phosphorylation of inhibitory kappa B alpha (IκBα), resulting in decreased TNF-α and increased IL-10 production by macrophages (Figure 3). These changes ameliorate lung inflammation and fibrosis [100].

(8) Microcystin-leucine arginine (LR): The binding of microcystin-LR to glucose-regulated protein 78 kDa (GRP78) suppresses endoplasmic reticulum unfolded protein response (UPR) signaling, which inhibits M2 macrophage polarization (Figure 3) and leads to attenuated EMT and fibroblast–myofibroblast transition (FMT). As a result, microcystin-LR exerts a therapeutic effect on lung fibrosis [101].

(9) Niclosamide: S100A4 is secreted by M2 macrophages to enhance collagen production in lung fibroblasts. The inhibition of S100A4 production with niclosamide reduces collagen deposition and attenuates lung fibrosis (Figure 3) [75].

(10) Nintedanib: By inhibiting CSF1R, nintedanib blocks the M2 differentiation of monocytes (Figure 3), which reduces macrophage activation and ameliorates fibrosis in a CTD-ILD model [102].

(11) Pirfenidone: Pirfenidone has an indirect inhibitory effect on fibroblast proliferation by suppressing macrophage polarization toward the M2 phenotype (Figure 3), which is therefore helpful for lung fibrosis [103].

(12) PRI-724: β-catenin signaling promotes the TGF-β expression of macrophages, which contributes to collagen synthesis in fibroblasts. As a result, the β-catenin pathway inhibitor PRI-724 inhibits the production of TGF-β by alveolar macrophages and ameliorates collagen deposition in fibrotic lungs (Figure 3) [79,104].

(13) Recombinant human pentraxin-2 (rhPTX-2): Pentraxin-2, also called serum amyloid P (SAP), reduces M2 differentiation and subsequent lung fibrosis (Figure 3) [80,81]. In line with this, rhPTX-2 prevents the deterioration of lung function in PF patients [105].

(14) Schisandra: Schisandra decreases TGF-β, reduces the levels of Smad3 and Smad4, and upregulates Smad7, suppressing M2 macrophage polarization (Figure 3). Accordingly, Schisandra exerts protective effects on lung fibrosis [106].

(15) Ruxolitinib: In CTD-ILD, ruxolitinib, which inhibits JAK/STAT, prevents the upregulation of proinflammatory molecules (CXCL10, IFI44, NOS2, and TNF-α) and profibrotic molecules (ARG1, CHI3L3, FLT1, IL4Ra, and RETNLA) (Figure 3). Ruxolitinib concomitantly reduces M2 polarization and the mononuclear pulmonary infiltrate, leading to significantly reduced CTD-ILD in an animal model [107].

(16) Tacrolimus: In the animal model of PF, tacrolimus inhibits signaling of JAK/STAT pathways and suppresses M2 polarization and M2-induced myofibroblast activation, thereby alleviating PF progression [108].

## 6. Investigational Agents in Preclinical and Clinical Trials of CTD-ILD

In practice, there is no standard treatment protocol for CTD-ILD. However, considering the autoimmune nature of CTDs, immunomodulatory drugs are the cornerstones in the pharmacologic treatment of CTD-ILD [109]. Considering these uncertainties, therapies that address the ILD component of CTD are currently a field of active research [110]. According to clinicaltrials.gov (accessed on 25 March 2023), 39 clinical trials exploring the therapeutic effects of various agents for CTD-ILD have been planned or are already underway or completed (Table 3). Of the 16 therapeutic agents that act on non-macrophage cells, three (Cyclophosphamide, Mycophenolate Mofetil, and Tadalafil) exhibit therapeutic effects. Of the four therapeutic agents which act on macrophages, three (Nintedanib, Pirfenidone, and Tacrolimus) exhibit therapeutic effects, which is borderline higher than the percentage of agents which act on non-macrophage cells (*p* = 0.06 by Fisher’s Exact Test). However, most available clinical trials examined agents which act on non-macrophage cells (25 clinical trials) rather than agents that act on macrophage cells (14 clinical trials). Therefore, it is hoped that future clinical trials will focus more on macrophage-directed agents.

## 7. Concluding Remarks and Future Directions

Despite the significant morbidities and mortalities caused by CTD-ILD, publications detailing how macrophages contribute to ILD in the context of CTD are scarce. More research is necessary to elucidate the distinct roles of specific lung macrophage populations in the CTD-ILD, with the aim of providing new insights into the development of macrophage-directed therapeutic targets and diagnostic tools [31]. Current knowledge is mostly extrapolated from findings on animal models of PF. However, well-established mouse macrophage counterparts might not always be available in humans. A humanized mouse model in which mice are transplanted with alveolar organoids and hematopoietic stem and progenitor cells could be exploited to overcome these issues [69]. Blocking macrophage molecular mechanisms, boosting macrophage signaling pathways, or combinations of these may prove fruitful and provide opportunities to relieve the paucity of proven effective treatments [137] and meet the clinical needs of CTD-ILD. The literature shows that aiming at specific cell types is a viable choice for several disease entities [138,139,140], especially CTDs [141,142]. Several systemic reviews also suggest the therapeutic potential of targeting specific cells in CTD-ILD [143,144]. Moreover, the pathogenesis of CTD-ILD involves other cell types (such as lymphocytes and fibroblasts) whose crosstalk with macrophages requires further characterization [74]. Deciphering these interactions could be another direction of research. We hope that our efforts to elucidate the pathophysiology of ILD macrophages in the context of CTD will offer new insights into the development of macrophage-directed therapeutic approaches for this group of debilitating diseases.

## Figures and Tables

**Figure 1 ijms-24-11995-f001:**
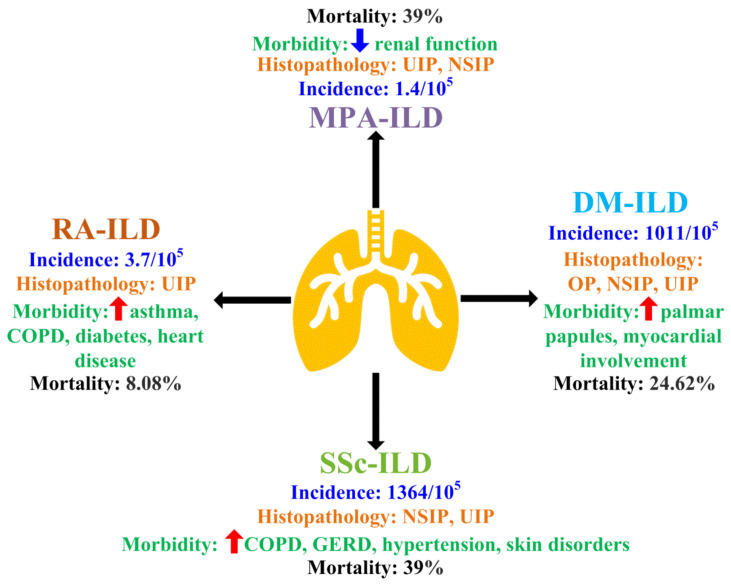
Incidence, histopathology features, morbidities, and mortalities of individual CTD-ILD discussed in this review. The figure summarizes incidence, histopathology features, morbidities, and mortalities of CTD-ILD discussed in this review. (1) The incidence of DM-ILD is 1011/10^5^. Histopathological changes associated with DM-ILD are organizing pneumonia (OP), nonspecific interstitial pneumonia (NSIP), and usual interstitial pneumonia (UIP). Patients have increased palmar papules and myocardial involvement, with mortality rate of up to 24.62%. (2) The incidence of MPA-ILD is around 1.4/10^5^. The pathological patterns are often UIP and NSIP. It is associated with decreased renal function and has a mortality rate of up to 39%. (3) The estimated incidence of RA-ILD is 3.7/10^5^. It is characterized by UIP. The patients usually suffer from increased asthma, chronic obstructive pulmonary disease (COPD), diabetes, and heart disease, and the mortality rate is 8.08%. (4) The incidence of SSc-ILD is 1364/10^5^. It is characterized by NSIP and UIP, accompanied by COPD, gastroesophageal reflux disease (GERD), hypertension, and skin disorders and a mortality rate of about 39%. Red arrow—increase; blue arrow—decrease.

**Figure 2 ijms-24-11995-f002:**
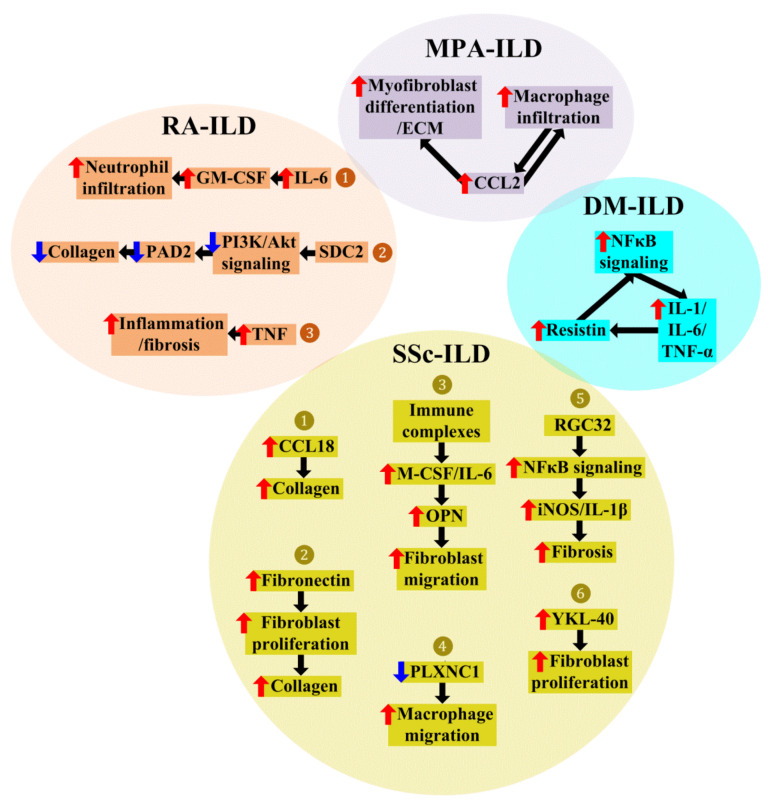
Roles of macrophages in CTD-ILD. The figure depicts mechanisms through which macrophages modulate CTD-ILD. In DM-ILD, macrophages express resistin. Resistin activates the NFκB signaling pathway, increasing the secretion of IL-1, IL-6, and TNF-α. This triggers the expression of resistin in macrophages, creating a self-perpetuating reaction to mediate the inflammatory pathogenesis of DM-ILD. In MPA-ILD, macrophages secrete CCL2, which attracts more macrophages to infiltrate. The recruited macrophages produce CCL2, initiating a self-propagating response. Meanwhile, CCL2 stimulates fibroblast differentiation into myofibroblasts and generates ECM, leading to exaggerated lung fibrosis. In RA-ILD, macrophages are implicated in three ways: (1) Exaggerated IL-6 secretion by macrophages triggers the proliferation of GM-CSF-producing T cells, and GM-CSF recruits neutrophil infiltration into the lung, leading to RA-ILD progression. (2) Macrophage-derived SDC2 interferes with PI3K/Akt signaling, decreases PAD2 in fibroblasts, and reduces collagen deposition in the lungs. (3) Macrophages produce high levels of TNF, which leads to an inflammatory phase predominated by cellular infiltration into the lungs and a subsequent shift to a fibrotic phase with irreversible collagen deposition. In SSc-ILD, six pathways are activated by macrophages to drive SSc-ILD: (1) Macrophages produce increased CCL18, which facilitates collagen production. (2) Macrophages release increased fibronectin, which induces fibroblast proliferation and collagen deposition. (3) Immune complexes stimulate the production of M-CSF and IL-6, which further induce OPN from monocytes. In turn, OPN triggers fibroblast migration, driving fibrosis progression. (4) SSc-ILD shows decreased PLXNC1 expression in macrophages, which boosts macrophage migration. (5) RGC32 promotes iNOS/IL-1β-directed inflammation through NFκB signaling and subsequently enhances the fibrotic response. (6) Macrophage-derived YKL-40, which is increased in SSc-ILD, promotes fibroblast proliferation. Red arrow—increase; Blue arrow—decrease; Black arrow—lead to.

**Figure 3 ijms-24-11995-f003:**
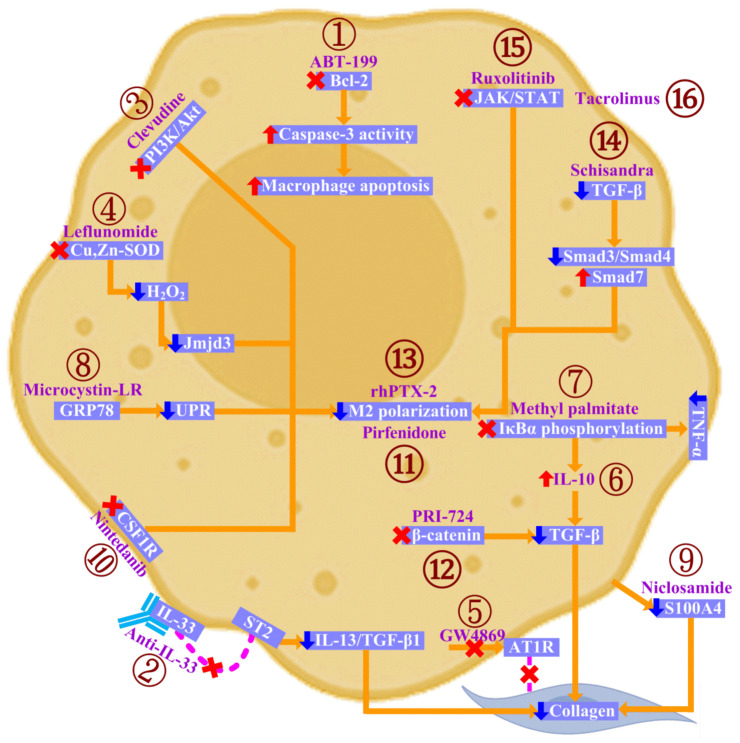
Potential agents to target macrophages in CTD-ILD. The figure depicts potential strategies to target macrophages in the context of CTD-ILD according to the literature. (1) ABT-199 inhibits Bcl-2. As a result, ABT-199 augments caspase-3 activity and enhances macrophage apoptosis, counteracting lung fibrosis. (2) Anti-IL-33 blocks IL-33 action on ST2 and thereby downregulates IL-13 and TGF-β1 production, which decreases collagen synthesis by fibroblasts. (3) Clevudine, as an inhibitor of the PI3K/Akt signaling pathway, prevents M2 polarization and the subsequent fibrotic response of the lungs. (4) By inhibiting Cu,Zn-SOD-mediated H_2_O_2_ generation, leflunomide downregulates Jmjd3 expression, lowers M2 polarization, and diminishes lung fibrosis. (5) GW4869, which eliminates the secretion of AT1R-carrying exosomes from macrophages, decreases collagen synthesis in fibroblasts and ameliorates lung fibrosis. (6) IL-10 curbs TGF-β production in macrophages and prevents collagen deposition by fibroblasts. (7) Methyl palmitate blocks IκBα phosphorylation, reducing TNF-α and boosting IL-10 expression in macrophages, which inhibit lung inflammation and fibrosis. (8) After binding to GRP78, microcystin-LR interferes with UPR signaling, subsequently preventing M2 macrophage polarization and leading to attenuated lung fibrosis. (9) Niclosamide interferes with S100A4 production by macrophages, which ameliorates collagen production by fibroblasts. These changes culminate in reducing the fibrosis of the lungs. (10) Nintedanib limits M2 differentiation by blocking CSF1R and consequently improves fibrosis in a CTD-ILD model. (11) Pirfenidone acts on macrophages to block M2 polarization and, as a result, suppresses fibroblast proliferation and restricts lung fibrosis. (12) The β-catenin pathway inhibitor PRI-724, which blocks β-catenin signaling, lowers TGF-β production in macrophages and contributes to mitigated collagen production by fibroblasts. (13) rhPTX-2 prevents M2 differentiation and avoids PF deterioration. (14) Schisandra reduces TGF-β and downstream Smad3/Smad4 while stimulating Smad7 production. These signaling events disrupt M2 polarization and thereby improving lung fibrosis. (15) Ruxolitinib and (16) Tacrolimus, by inhibiting JAK/STAT signaling, suppress M2 polarizations and hence improve lung fibrosis in an animal model. Red arrow—increase; blue arrow—decrease; cross symbol—block; dashed line—act on.

**Table 3 ijms-24-11995-t003:** Efficacy of investigational agents in clinical trials of CTD-ILD in clinicaltrials.gov.

Agents	Targeted Cells	Diseases	Study	Outcome ^a^
Abatacept	T cells [111]	Antisynthetase syndrome-associated ILD	NCT03215927	Active
Abatacept	T cells [111]	RA-ILD	NCT03084419	Unknown
Abituzumab	Epithelial cells [112]	SSc-ILD	NCT02745145	Terminated due to slow enrollment [112]
Basiliximab	Lymphocytes [113]	DM-ILD	NCT03192657	Unknown
Belimumab	B cells [114]	SSc-ILD	NCT05878717	Not yet recruiting
Bortezomib	Fibroblasts [115]	SSc-ILD	NCT02370693	Completed
Bosentan	Fibroblasts [116]	SSc-ILD	NCT00070590	Ineffective [117]
Bosentan	Fibroblasts [116]	SSc-ILD	NCT00319033	Completed
Cyclophosphamide	Lymphocytes [118]	Antisynthetase syndrome-associated ILD	NCT03770663	Recruiting
Cyclophosphamide	Lymphocytes [118]	SSc-ILD	NCT00883129	Improve [119]
Cyclophosphamide	Lymphocytes [118]	SSc-ILD	NCT01570764	Completed
Cyclosporin A	T cells [120]	Sjogren’s syndrome-associated ILD	NCT02370550	Unknown
Ixazomib	Fibroblasts [121]	SSc-ILD	NCT04837131	Recruiting
Mycophenolate mofetil	Lymphocytes [122]	Myositis-associated ILD	NCT05129410	Recruiting
Mycophenolate mofetil	Lymphocytes [122]	SSc-ILD	NCT00883129	Improve [119]
Mycophenolate mofetil	Lymphocytes [122]	SSc-ILD	NCT02896205	Ineffective [123]
Mycophenolate mofetil	Lymphocytes [122]	SSc-ILD	NCT05785065	Not yet recruiting
N-acetylcysteine	Macrophages [124]	CTD-ILD	NCT01424033	Terminated due to departure of principal investigator.
Nintedanib	Macrophages	Myositis-associated ILD	NCT05335278	Recruiting
Nintedanib	Macrophages	Myositis-associated ILD	NCT05799755	Not yet recruiting
Nintedanib	Macrophages	SSc-ILD	NCT02597933	Effective [125]
Nintedanib	Macrophages	SSc-ILD	NCT03313180	Completed
Pirfenidone	Macrophages	CTD-ILD	NCT04928586	Recruiting
Pirfenidone	Macrophages	CTD-ILD	NCT05505409	Recruiting
Pirfenidone	Macrophages	DM-ILD	NCT02821689	Unknown
Pirfenidone	Macrophages	DM-ILD	NCT03857854	Unknown
Pirfenidone	Macrophages	RA-ILD	NCT02808871	Slow FVC decline [126]
Pirfenidone	Macrophages	SSc-ILD	NCT03221257	Completed
Pirfenidone	Macrophages	SSc-ILD	NCT03856853	Unknown
Pomalidomide	T cells [127]	SSc-ILD	NCT01559129	Discontinued [127]
PRA023	Fibroblasts [128]	SSc-ILD	NCT05270668	Recruiting
Rituximab	B cells [129]	CTD-ILD	NCT01862926	Not superior to cyclophosphamide [129]
Rituximab	B cells [129]	RA-ILD	NCT00578565	Inconclusive
Tacrolimus	Macrophages	Myositis-associated ILD	NCT00504348	Improve [130]
Tacrolimus	Macrophages	Myositis-associated ILD	NCT02159651	Improve [131]
Tadalafil	Fibroblasts [132]	SSc-ILD	NCT01553981	Improve [133]
Tofacitinib	Myeloid derived suppressor cells [134]	RA-ILD	NCT04311567	Recruiting
Tofacitinib	Myeloid derived suppressor cells [134]	RA-ILD	NCT05246293	Recruiting
Vixarelimab	Fibroblasts [135,136]	SSc-ILD	NCT05785624	Recruiting

^a^ Recorded at clinicaltrials.gov or in the literature. FVC: functional vital capacity. Unknown: no result has been posted, or the recruitment status is unknown.

## Data Availability

No new data were created or analyzed in this study. Data sharing is not applicable to this article.

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
