# Peer review of "The Role of Macrophages in Connective Tissue Disease-Associated Interstitial Lung Disease: Focusing on Molecular Mechanisms and Potential Treatment Strategies"

_ijms, 2023, doi:10.3390/ijms241511995_

Round 1
Reviewer 1 Report
Summary:
The review entitled, Macrophages in Connective Tissue Disease-associated Interstitial Lung Disease (CTD-ILD): Molecular Mechanisms and Potential Treatment Strategies discuss the characteristics, functions and origins of subsets of macrophages involved in CTD-ILD, including resident alveolar, interstitial and monocyte-derived macrophages. The work's strengths are the evidence in favor of a role in macrophages in CTD-ILD progression and might be a potential target to inhibit CTD-ILD and prolong the survival of affected patients. However, I found the evidence of a specific role for macrophage function to be very weak. This work needs a significant reorganization before publication.
I recommend that the authors should address all the significant concerns. As a researcher, these sections would be very useful for all the readers. Authors need to revise title.
1. Authors need to rewrite the abstract and summarize that what he is discussing in the whole review and how the authors review provide some new information. Need justification.
2. From section 2 and 3 authors discussed the relationship between macrophages and pulmonary fibrosis and ILD. There is a published article on a similar topic, but the authors have discussed much more versatile way each topic and also included new findings in their review.
a. Byrne, Adam J., Toby M. Maher, and Clare M. Lloyd. "Pulmonary macrophages: a new therapeutic pathway in fibrosing lung disease?" Trends in molecular medicine 22, no. 4 (2016): 303-316.
b. Ogawa, Tatsuro, Shigeyuki Shichino, Satoshi Ueha, and Kouji Matsushima. "Macrophages in lung fibrosis." International Immunology 33, no. 12 (2021): 665-671.
c. Gu, Yanrong, Toby Lawrence, Rafeezul Mohamed, Yinming Liang, and Badrul Hisham Yahaya. "The emerging roles of interstitial macrophages in pulmonary fibrosis: A perspective from scRNA-seq analyses." Frontiers in Immunology 13 (2022).
3. In section 5. “Targeting macrophages in CTD-ILD” All the targets/inhibitors are associated with lung fibrosis. Therefore, the authors are also requested to include a section (with a table) to write on the investigational agents that are currently in preclinical and clinical trial and/or reported against different CTD-ILD their stage of trials, their outcome, to make this review more future directive.
4. In Section 6. “Concluding remarks” authors mentioned that a significant morbidities and mortalities caused by CTD-ILD. The authors are requested to include a figure on the CTD-ILD mortality/morbidities and incidence from the available human dataset.
5. In a few instances, the spaces between adjacent words are missing. There are also grammatical mistakes. Please check the spelling and grammar. I would suggest that the whole manuscript be thoroughly revised to improve clarity and readability.
6. Reference section need revisions and authors need to cite more recent papers.
In a few instances, the spaces between adjacent words are missing. There are also grammatical mistakes. Please check the spelling and grammar. I would suggest that the whole manuscript be thoroughly revised to improve clarity and readability.
Reviewer 2 Report
In this manuscript, Tseng et al. has reviewed the biological function of macrophages in pathogenesis and progression of connective tissue disease-associated interstitial lung disease (CTD-ILD). The authors have provided a comprehensive summary of the molecular mechanisms including both signaling pathways and secretory product underlying which macrophages play a role in disease progression of CTD-ILD. Importantly, the therapeutic strategies that target macrophages in treatment of CTD-ILD have been discussed in the review. Overall, the authors have covered different aspects of the impact of macrophages in CTD-ILD as well as the implications for development of clinical treatment. However, more information is required for the pathology / clinical aspects of ILD and the section for molecular mechanisms of macrophages in ILD needs to be re-organized. In addition, the language needs editing.
Specific comments:
1) In Introduction section, more information is required about the clinical diagnosis, pathology features as well as mortality rate of CTD-ILD.
2) In Section 2, the molecular mechanisms underlying which macrophages affect pathogenesis of CTD-ILD need to be organized in a signaling pathway level instead of listing each individual molecule.
3) In Section 4, what’s DM-ILD, MPA-ILD, RA-ILD and SSc-ILD, and what are the relationship regarding CTD-ILD?
4) In section 5, what’s the standard treatment for CTD-ILD? What are the additional advantages targeting macrophages compared to other therapeutic methods?
5) What’s the current research trend and future insights for the impact of macrophages in CTD-ILD?
The language needs editing in grammar.
